# Seasonal Variability of *Juniperus communis* L. Berry Ethanol Extracts: 2. In Vitro Ferric Reducing Ability of Plasma (FRAP) Assay

**DOI:** 10.3390/molecules27249027

**Published:** 2022-12-18

**Authors:** Jozef Fejér, Daniela Gruľová, Adriana Eliašová, Ivan Kron

**Affiliations:** 1Faculty of Humanities and Natural Sciences, Department of Ecology, University of Presov, 17. Novembra 1, 08001 Presov, Slovakia; 2Training & Consulting Ltd., 01001 Žilina, Slovakia

**Keywords:** ethanol extracts, antioxidant activity, FRAP, terpene hydrocarbons, total phenols

## Abstract

In the present study, the seasonal variability of the in vitro ferric reducing ability of plasma (FRAP), total phenols, and terpene hydrocarbon content in 70% ethanol extracts were evaluated. The samples of crushed (CBs) and non-crushed ripe juniper berries (NCBs) collected at five localities in North-East Slovakia during the years 2012–2014 were compared. The method of preparation of the extract influenced the amount of dry matter (DM) in the extracts. In the CB extracts were statistically higher contents of DM (from 13.91 ± 0.11 g·L^−1^ to 23.84 ± 0.14 g·L^−1^) compared to NCB extracts (from 1.39 ± 0.01 g·L^−1^ to 16.55 ± 0.09 g·L^−1^). The differences in antioxidant activity between the investigated localities were statistically significant for both types of extract. For example, in 2013 in the locality of Zbojné, the FRAP in NCBs was 76.62 µmol·L^−1^·g^−1^ DM and in CBs was 138.27 µmol·L^−1^·g^−1^ DM, while in the Miľpoš locality, in NCBs there was 232.66 µmol·L^−1^·g^−1^ DM and in CBs there was 1178.98 µmol·L^−1^·g^−1^ DM. The differences in the antioxidant activity between the studied years in the case of NCB extracts were not statistically significant. In the case of CB extracts, significant differences between the evaluated years were found. Statistics by ANOVA confirmed that CB extracts prepared from berries in the year 2013 showed significantly higher activity compared to CB extracts from berries from the years 2012 and 2014. Based on the Pearson we found a negative correlation coefficient between the FRAP assay and the content of total polyphenols in NCB extracts (−0.531 in 2012; −0.349 in 2013; and −0.224 in 2014). In contrast, CB extracts showed a positive correlation coefficient (0.843 in 2012; 0.742 in 2013; 0.617 in 2014).

## 1. Introduction

Free radicals are formed during normal cell metabolism [1]. An imbalance between oxidants and antioxidants in favor of oxidants creates oxidative stress [2], which is the cause of various diseases [3,4]. The balance between antioxidation and oxidation is considered a fundamental concept for a healthy biological system [5]. The damage caused by free radicals can be repaired by a class of molecules generally referred to as antioxidants [4]. A biological antioxidant is defined as any antioxidant or reducing substance which, when present in low concentrations of an oxidizable substrate, significantly slows down or prevents the oxidation of this substrate [6]. There are several methods for evaluating total antioxidant activity or scavenging of free radicals [7]. The FRAP method is based on the ability of antioxidants to reduce iron. The principle of this method is based on the reduction of the complexity of ferric iron and 2,3,5-triphenyl-1,3,4-triaza-2-azoniacyclopenta-1,4-diene chloride (TPTZ) to the ferrous form at low pH. This reduction is measured on a spectrophotometer at 592 nm absorption. By comparing the absorption change in the test mixture with those obtained from increasing concentrations of Fe^3+^ and expressed as mM of Fe^2+^ equivalents per kg or per L of sample, we can obtain the FRAP values [8]. Plants have been used for thousands of years as medicines for treating a variety of different diseases and medical complaints by most civilizations [9]. Plant extracts contain numerous bioactive compounds, such as alkaloids, terpenes, and polyphenols [4], which have antioxidant effects [9]. *Juniperus communis* L. is an evergreen dioecious gymnosperm shrub [9,10,11,12], with a broad ecological amplitude adapted to various climatic conditions. It is distributed throughout the world and, due to this wide geographical distribution, there are many remarkable differences in morphological characteristics and the chemical composition of secondary metabolites [9,12,13,14]. The wide geographical distribution is the principal reason for the remarkable variation in the morphological characteristics and secondary metabolites’ chemical compositions [14]. The content and composition of phenolics found in *J. communis* L. species depend on genotype, plant part, origin, age, gender, and solvent used to extract phenolics and perform the studies but, in a general way, they increase with latitude and plant age [11,15,16]. The differences in the content of phenolic compounds between male and female leaves and berries have also been found [17]. The astringent blue-black seeds are commonly too bitter to eat raw and are dried for their use as a culinary component in different regions of the world. The dried berries are crushed or grounded to release their flavor before these are added to a dish. These are used to flavor meat, soups, sauces, stews, stuffing, and pickled foods [9]. Juniper berries are used in traditional medicine [18], as well as used to produce alcoholic beverages such as flavor in gin, liquors, bitters, Swedish beer, and borovička, the Slovak national alcoholic beverage similar to gin [19]. The branches and berries of juniper are burnt in temples to purify air during religious ceremonies [20]. The supplementation of berries or their essential oil experimentally has been observed to have a positive impact on performance and yield in quails, and such supplementation has been found to be a better alternative for synthetic antioxidants for the preservation of meat [9]. This study builds on previous research that has been published by [14]. In this study, we focused on the evaluation of the efficiency of a 70% ethanol extract of non-crushed (NCBs) and crushed juniper ripe berries (CBs), obtained from the natural populations of north-east Slovakia within three years, on the antioxidant activity by the method of in vitro Ferric reducing ability of plasma (FRAP) assay. Based on our knowledge, it is the first study of antioxidant activity by the FRAP method of extracts from *J. communis* galbuls. It is well known that the environmental conditions have an influence on the development of the natural components in plant matrices. The aim of the study is to compare the quality and the potential antioxidant activity of the extracts in different localities and different years.

## 2. Results

### 2.1. Climatic Conditions within Studied Three Years in Five Localities

Base climatic parameters such as precipitation and temperatures were compared within the five localities and three studied years. The data were collected from the nearest meteo stations to the locality from the official web page of the Slovak hydrometeorology department (www.shmu.sk; accessed on 25 October 2014). In 2012, the highest average precipitation was measured in the Zbojné locality (808 mm) and the lowest in the Hôrka locality (550 mm) in the Presov self-governing region. A similar observation was made in 2013, when the highest precipitation was in Zbojné (815 mm) and the lowest in Hôrka (676 mm). The changes were noted in third year of research in 2014, when the highest precipitation was in two localities, Miľpoš and Lačnov (880 mm), following the Kamienka locality (836 mm), and the Hôrka (774 mm) and Zbojné (752 mm) localities noted the lowest amount of precipitation. Generally, in the first year of our research, the lowest amount of precipitation in all five localities (691 mm) was noted, while the highest amount of precipitation was noted in the third year, 2014 (810.5 mm) (Figure 1).

Average temperatures in different localities were observed as follows. In 2012, the average highest temperature was noted in the Zbojné locality (8.3 °C) and the lowest in the Hôrka locality (6.9 °C). As was described in the precipitation, an identical pattern is noted in temperature; in 2013, again the highest average temperature was noted in Zbojné (8.6 °C) and the lowest in Hôrka (7.0 °C). A similar pattern continues also in the third year, 2014, when in Zbojné was the highest average temperature (9.5 °C) and the lowest was in Hôrka (8.2 °C) (Figure 2). There was a significant increase of the average temperature and precipitation between the first and third years of our research in PSGR in general. While in 2012, the average precipitation was 691 mm; in 2013, it was 739 mm; and in 2014, it was already 810.5 mm. The average temperature follows the pattern of precipitation. In 2012, the average temperature in PSGR was 7.4 °C; in 2013, it was 7.6 °C; and in 2014 it was noted as 8.8 °C.

### 2.2. Content of Phenols and Terpene Hydrocarbons in Juniper Extracts

The evaluation of phenol content in non-crushed juniper berries (NCBs) and crushed juniper berries (CBs), as well as terpene hydrocarbon content in crushed berries, were evaluated for each locality and for three years. Generally, the higher number of phenols was noted in the extracts from non-crushed juniper berries. Each year, the significant differences were evaluated between localities (Table 1). In 2012, the highest content of phenols was noted in extracts from the Zbojné locality (36.89 GAE mg.g^−1^ DM) and the lowest in the Hôrka locality (8.49 GAE mg.g^−1^ DM) in NCBs. In the same year, the highest amount of phenols in CBs was noted in extracts from Hôrka (18.77 GAE mg.g^−1^ DM), and the lowest was from extracts from Miľpoš (9.82 GAE mg.g^−1^ DM). The rest of the three localities noted a similar amount (11.13–11.67 GAE mg.g^−1^ DM). In the second year (2013) of our experiment, the highest content of phenols in NCBs was evaluated in extracts from Hôrka (20.41 GAE mg.g^−1^ DM) and the lowest in extracts from lačnov (9.12 GAE mg.g^−1^ DM). In the extracts from CBs, a higher content of phenols was evaluated in an almost similar amount in three localities—Kamienka, Miľpoš, Lačnov (11.41, 10.88 and 10.84 GAE mg.g^−1^ DM)—and, at the lower end, almost the same content was evaluated in Hôrka and Zbojné (7.11 and 7.84 GAE mg.g^−1^ DM). In the last year of our observation, the highest amount of phenols in NCBs was evaluated in the Hôrka locality (42.23 GAE mg.g^−1^ DM) while the lowest was in the Zbojné locality (14.97 GAE mg.g^−1^ DM). Three other localities had similar contents of phenols: Kamienka (20.6 GAE mg.g^−1^ DM), Lačnov (21.64 GAE mg.g^−1^ DM), and Miľpoš 23.4 GAE mg.g^−1^ DM). In the extracts from CBs, the highest amount of phenols was evaluated in extracts from Lačnov (12.53 GAE mg.g^−1^ DM), and a similar amount was also measured in two other localities: Miľpoš (11.58 GAE mg.g^−1^ DM) and Hôrka (10.7 GAE mg.g^−1^ DM). A lower amount of phenols were evaluated in Zbojné and Kamienka (8.92 and 6.87 GAE mg.g^−1^ DM).

### 2.3. Ferric Reducing Ability of Plasma (FRAP) Assay

The results of antioxidant activity and dry matter for extracts of non-crushed (NCB) and crushed berries (CB) are shown in Table 2. Depending on the locality and the year of juniper berries collection, the amount of dry matter of NCB extracts varied considerably. Even though the berries were naturally dried under the same conditions after harvesting, probably the differences in environmental conditions at the localities and the years influenced the thickness of the skin and thus its permeability to substances. As for the dry matter content in the CB extracts, it was significantly higher compared to the NCB extracts.

The number of extracted substances subsequently influenced the antioxidant activity of individual extracts, which was significantly higher in CB extracts compared to NCB extracts (Table 3). Generally, there are differences in ferric reducing ability of plasma assay between the extracts from NCB and CB. The higher potential vas evaluated in CB extracts. In 2012. At CB extracts, statistically significant differences in antioxidant activity were found between the studied localities (*p* < 0.001) as well as between the studied years (*p* < 0.001). Statistics by ANOVA confirmed that CB extracts prepared from berries in the year 2013 showed significantly higher activity compared to CB extracts from berries from the years 2012 and 2014.

### 2.4. Correlation Analysis

We have evaluated a negative correlation coefficient between the antioxidant activity assayed by the FRAP (µmol.L^−1^.g^−1^ DM) method and the content of total polyphenols (GAE mg.g^−1^ DM) in NCB ethanol extracts. In 2012, there was a correlation coefficient r = −0.531, in 2013 r = −0.349 and in 2014 r = 0.224. Opposite to these results, in CB extracts a positive correlation coefficient between antioxidant activity and total polyphenols was evaluated. In 2012, it was r = 0.843; in 2013, it was r = 0.742; and in 2014, it was r = 0.617. Correlation coefficients were decreasing in the research years from 2012 to 2014. Correlation coefficients between antioxidant activity and the content of terpene hydrocarbons ranged from r = 0.504 in the year 2012, until negative coefficient in 2013 (r = −0.684) and in 2014 (r = −0.220) (Table 3).

## 3. Discussion

The *J. communis* plant is not only a rich source of nutrition but also is rich in aromatic oils and their concentration varies in different parts of the plant (berries, leaves, aerial parts, and root). The fruit berries contain essential oil (0.5% in fresh and 2.5% in dry fruit), invert sugars (15–30%), resin (10%), catechin (3–5%), organic acid, terpenic acids, leucoantho-cyanidin besides a bitter compound (Juniperine), flavonoids, tannins, gums, lignins, wax, etc [21,22,23]. Junipers produce nearly 580 secondary metabolites, including anticancer lignans (podophyllotoxin and other derivatives), sesquiterpenes, diterpenes (more than 220 structures), flavonoids, etc. Polyphenols are a large group of compounds, comprising flavonoids, stilbenes, polyphenolic acids, lignans, etc. These substances exhibit various biological activities, some of them with poorly understood mechanisms [9,11]. The total polyphenol content of plant extracts correlates with their antioxidant activity [11]. Generally, the majority of phenolics reported in *J. communis* L. plant parts include 5-Ocaffeoylquinic and quinic acids, catechin, epicatechin, amentoflavone, quercetin, luteolin, apigenin, and naringenin and their derivatives [12]. The total phenolic content in ripe berries varied in the range from 0.19–99 mg equivalent of gallic acid (GAE) per g dry weight (dw) of *J. communis* in as different origins as Turkey, Slovakia, Canada, Australia, Portugal, and Romania [16,24,25,26,27,28,29]. A previous study reported a phenolic substances of 6.86 mg.g^−1^ GAE and an antioxidant activity of FRAP of 0.24 mmol Fe^2+^.g^−1^ in water extracts of juniper berries [30]. Aqueous extracts of juniper berries native to Egypt contained 1.01 mg GAE mL^−1^ of total phenolic and showed an antioxidant FRAP activity of 3.77 mM TE mL^−1^ [31]. In water extracts from juniper berries originating from Egypt, another study [32] found a total phenol content of 838.0 mg.100 g^−1^ DM and a FRAP activity of 6.82 mmol.100 g^−1^ DM. A high ferric reducing ability of plasma assays was reported [33] in methanolic and aqueous extracts of the unripe fruits of *Juniper oxycedrus*. A scientific study [34] evaluated the antioxidant activity of the essential oil of juniper berries by the FRAP method, which was 215.9 mmol.kg^−1^. For the components of the essential oil limonene and borneol, the activity was 37.4 mmol.kg^−1^ resp. 28.7 mmol.kg^−1^. The reducing abilities of the components of volatile oils are lower than those of volatile oils. In addition to the terpenes, other biologically active compounds may also contribute to ferric reduction and in electron scavenging. For example, for β-pinene, the FRAP value 6.5 µM Fe.mg^−1^ was found [35]. The antioxidant activity is influenced by the extraction agent used. The FRAP activity of 241.1 in the 50% ethanol extract, 229.7 in the 70% ethanol extract, and 246.3 in the water extract in mM Trolox equivalents per g dry plant extract was evaluated [36]. Another study [7] reported that an ethanol extract was found with the highest frequency for an antioxidant study. Through research, we found that the content of phenolic substances and antioxidant activity by the FRAP method varies depending on the locality, the year of collection, and the method of preparation of the extract (NCB and CB extracts).

The influence of abiotic stress on the content of total polyphenols and antioxidant activity is known. By testing two varieties of barley, it was confirmed that drought as an abiotic stress caused an increase in antioxidant activity compared to control samples. The same tendency was also observed with intensive irradiation of FAR = 800 µmol.m^−2^.s^−1^ [37]. The influence of the year of juniper fruit collection on the content of total phenolic substances and antioxidant activity against the hydroxyl radical is known [26]. The content of the essential oil and its composition are similarly affected. The influence of environmental conditions, namely temperature and precipitation, as well as humus content and pH of the soil, on the content of essential oil and its components has been demonstrated [14]. The number of extracted substances for individual types of extracts also depends on the amount of essential oil and its individual components. In addition to phenols, these can participate in the interaction with other substances contained in the extracts in the overall antioxidant activity. A high correlation coefficient between the antioxidant activity by the FRAP method and the total phenolic content of water extracts is reported as r = 0.966 [30] and r = 0.966 [32], respectively. A high correlation coefficient between the antioxidant activity by the FRAP method of ethanol extracts and total phenols of r = 0.903 was found by [36]. A study by [38] Stankov et al. (2020) reported a correlation coefficient of r = 0.768 for 70% ethanol extracts of ripe *Juniperus excelsa* berries. They found a negative correlation coefficient of r = −0.692 for berry extracts in 95% ethanol.

## 4. Materials and Methods

### 4.1. Plant Material

The juniper ripe berries were collected in five localities (Hôrka, Kamienka, Miľpoš, Lačnov, and Zbojné) in the Prešov self-governing region (PSGR) (Figure 3) which is geographically located in the north-east region of the Slovak republic. The collection was performed during September within the three years 2012–2014 and naturally dried. 

Precise locality characterization is described in a previous publication [14]. The sum of the precipitation (mm) and average temperatures (°C) in the localities within the observed three years are presented in Table 4.

### 4.2. Preparation of Extracts

Whole dried galbuli as non-crushed berries (NCBs) and berries crushed in a porcelain mortar as crushed berries (CBs) were macerated in 100 mL of 70% ethanol. The extractions were carried out with occasional stirring for 72 h at room temperature. The obtained extracts were filtered over a filter KA 1-M (very fast, Papírny Pernštejn, CZ). The dry matter (DM) content was determined by using filtrated extracts drying in Petri dishes in an oven at 105 °C for three hours, each extract in triplicate.

All chemicals used were of the highest quality: 70% ethanol (Centralchem, Bratislava chemical trading company, Slovakia). Double distilled water (DDW) was used for preparation of solutions. Absorbances of solutions of various test assays were determined in 1 cm quartz cells with a Shimadzu type UV-1800 spectrophotometer (Shimadzu, Japan).

### 4.3. Determination of Total Phenolic Content

The total phenolic content of the ethanol extracts of the leaves was determined with the Folin-Ciocalteu reagent (FCR, Merck) according to a procedure described by [39], with slight modifications [26,40,41]. Prior to the assay, the extracts were diluted with 70% ethanol to a ratio of 1:4 (*v/v*). Each diluted extract (0.1 mL) was sequentially well-mixed with 0.2 mL of FCR, 2 mL of DDW, and 1 mL of Na_2_CO_3_ saturated solution (20%, *w/v*, in water). After incubation in the dark at room temperature for 90 min, the absorbance of the reaction mixture was determined at 765 nm with a Shimadzu UV-1800 spectrophotometer. The blank sample was prepared in a similar way by replacing the extract with the same volume of 70% ethanol. The phenolic content was calculated as Gallic acid equivalents (GAE) per mL of extract based on a standard curve of Gallic acid (Merck). All the determinations were carried out four times for each extract. All the solutions were used on the day of preparation.

### 4.4. Determination of Terpene Hydrocarbons

The terpene hydrocarbon content of CB extracts was determined by gas chromatography with a Carlo Erba GC 6000 Vega Series 2 equipped with an ICU-600 programme controller, an EL 580 flame ionization detector (FID), a Spectra Physics SP 4270 integrator, and a SolGel-WAX GC Capillary Column (60 m × 0.25 mm i.d., film thickness 0.25 µm). An injector was heated to 200 °C, and a flame ionization detector was heated to 300 °C. The column temperature was maintained at 50 °C for 5 min and then programmed to increase to 200 °C at a rate of 5 °C/min and held at this temperature for 10 min; the injection volume was 1 µL; the split ratio was 1:20; nitrogen was used as a carrier gas (1.5 mL/min). The identification of monoterpene and sesquiterpene hydrocarbons was performed on the basis of the co-injection of some commercially available standards. Quantification was conducted with the peak area values obtained from GC-FID, using sabinene as an external standard. The content of terpene hydrocarbons was expressed as milligrams of sabinene equivalents per gram of the extract dry matter.

### 4.5. Ferric Reducing Ability of Plasma (FRAP) Assay

The working FRAP reagents and other reagents were prepared according to Benzie and Strain (1996). The only change was in the increase of hydrochloric acid concentration to 50 mmol L^−1^ for dissolving of 10 mmol L^−1^ TPTZ (2,4,6-tripyridyl-s-triazine). Aqueous solutions of iron (II) sulfate heptahydrate in the concentration range of 0–900 mmol L^−1^ were used for calibration at 600 nm (r = 0.9997). The assay was performed manually at room temperature. The solutions were pipetted into test tubes in duplicates. Solutions were mixed well and the absorbance at 700 nm was recorded after 5 min with the spectrophotometer. A Gallic acid solution with a concentration of 10 mmol L^−1^ was used for comparison. The FRAP of the samples (in µmol L^−1^) was calculated by the formula: FRAP = (Sample–Blank) × 500 (Standard–Blank). All the determinations of FRAP in the samples were performed at least four times. All the solutions were used on the day of preparation. The FRAP assay, which provides fast reproducible results, measures the ability of an antioxidant to reduce the ferric tripyridyltriazine (Fe+3-TPTZ) complex and produce the ferrous tripyridyltriazine (Fe+2-TPTZ) complex, which is blue in color. The details can also be found in our previous publications [41,42].

### 4.6. Statistical Analysis

The statistical software Statgraphics 5.0 (Statgraphics Technologies, Inc., The Plains, Virginia, USA) and multifactorial analysis of variance (ANOVA) method and an LSD (Least Significant Difference) of 95% were used for statistical analysis. Correlation analysis was performed by using the Excel software.

## 5. Conclusions

The study revealed relatively high antioxidant activity as determined by the FRAP method. Significantly higher activity was found in CB extracts compared to NCB extracts. It is related to the number of extracted substances, which was higher in CB extracts. Statistically significant differences in antioxidant activity by the FRAP method between the investigated localities could be caused by environmental conditions. These could affect the accumulation of secondary metabolites involved in antioxidant activity. Statistically significant differences in the activity of CB extracts between the evaluated years were also found. Differences in the correlation between antioxidant activity and total phenolic content compared to the literature were found. The differences between the results in this study and reports in the literature are probably due to the application of different extraction methods, methodologies for the detection of phenols, and the determination of antioxidant activity. Other substances, which were not determined in this research, probably also participated in the antioxidant activity. The study shows that there are natural products, which are safe for practical use, for example in food industry. Our findings shall be considered in the search for new products which will prolong the quality of specific food products or in prolonging their shelf life.

## Figures and Tables

**Figure 1 molecules-27-09027-f001:**
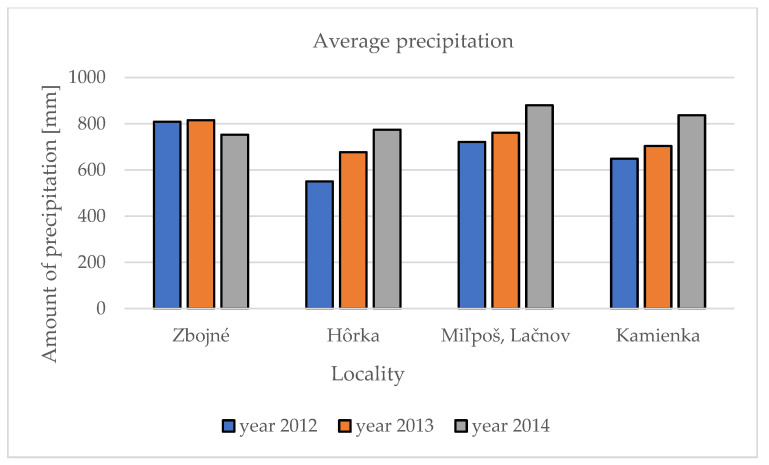
Average amount of the precipitation in five localities during three observation years.

**Figure 2 molecules-27-09027-f002:**
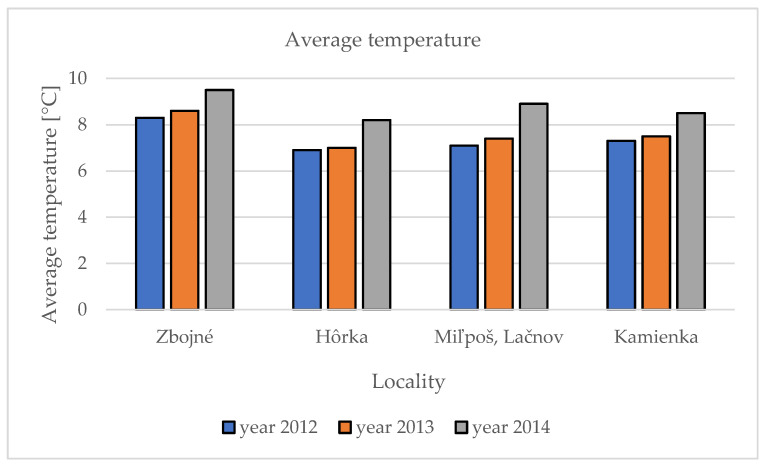
Average temperature in five localities during three observation years.

**Figure 3 molecules-27-09027-f003:**
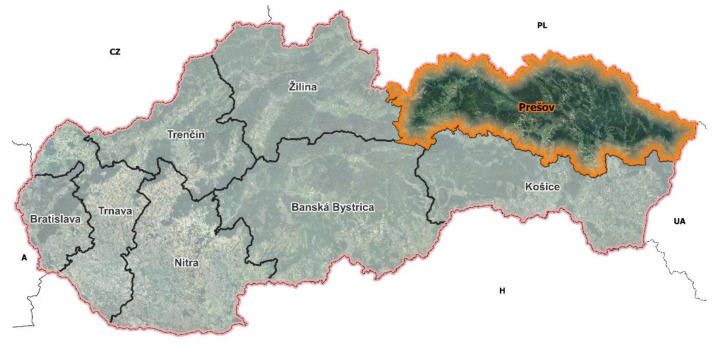
Prešov self-governing region marked in orange color (prepared by Dr. Štefan Koco).

**Table 1 molecules-27-09027-t001:** The content of phenols evaluated in NCBs and CBs and terpene hydrocarbons in CBs within three years.

		Phenols [GAE mg.g^−1 C^DM]		Terpene Hydrocarbon Content [mg.g^−1^ DM]
**Year**	**Locality**	**^A^NCB**	**^B^CB**	**CB**
2012	Zbojné	36.89 ± 0.31 ^c^	11.13 ± 0.80 ^b^	51.59 ± 8.15 ^b^
	Hôrka	8.49 ± 0.88 ^a^	18.77 ± 2.11 ^c^	64.70 ± 9.25 ^c^
	Miľpoš	10.47 ± 1.25 ^b^	9.82 ± 0.99 ^a^	38.68 ± 5.80 ^a^
	Kamienka	14.91 ± 0.75 ^b^	11.27 ± 0.85 ^b^	67.97 ± 7.50 ^c^
	Lačnov	30.32 ± 1.07 ^c^	11.67 ± 1.05 ^b^	62.92 ± 10.07 ^c^
**Year**	**Locality**	**NCB**	**CB**	**CB**
2013	Zbojné	14.5 ± 1.35 ^b^	7.84 ± 0.89 ^a^	69.93 ± 11.05 ^c^
	Hôrka	20.41 ± 0.89 ^c^	7.11 ± 1.01 ^a^	60.83 ± 8,70 ^b^
	Miľpoš	15.1 ± 1.54 ^b^	10.88 ± 1.26 ^b^	47.2 ± 7.08 ^a^
	Kamienka	12.3 ± 0.95 ^b^	11.41 ± 0.74 ^b^	59.17 ± 6.51 ^b^
	Lačnov	9.12 ± 0.60 ^a^	10.84 ± 1.20 ^b^	45.99 ± 7.36 ^a^
**Year**	**Locality**	**NCB**	**CB**	**CB**
2014	Zbojné	14.97 ± 1.12 ^a^	8.92 ± 0.83 ^a^	57.09 ± 9.02 ^a^
	Hôrka	42.23 ± 2.61 ^c^	10.7 ± 1.33 ^b^	79.33 ± 11.34 ^b^
	Miľpoš	23.4 ± 1.97 ^b^	11.58 ± 1.81 ^b^	64.97 ± 9.75 ^a^
	Kamienka	20.6 ± 1.67 ^b^	6.87 ± 0.47 ^a^	73.62 ± 8.10 ^b^
	Lačnov	21.64 ± 0.72 ^b^	12.53 ± 1.60 ^b^	61.85 ± 9.90 ^a^

^A^NCB–non-crushed berries; ^B^CB–crushed berries; ^C^DM–dry mass; the phenolic content was calculated as Gallic acid equivalents (GAE) per mL of extract on the basis of a standard curve of Gallic acid; Statistical differences (ANOVA, *p* < 0.001) between localities within each year are marked with different letters (^a,b,c^).

**Table 2 molecules-27-09027-t002:** Ferric reducing ability of plasma (FRAP) assay and the dry matter content in 70% ethanol extracts juniper berries.

		FRAP (µmol.L^−1^.g^−1 C^DM)		DM (g.L^−1^)	
**Years**	**Locality**	**^A^NCB**	**^B^CB**	**NCB**	**CB**
2012	Zbojné	49.50 ± 1.71 ^a^	282.57 ± 11.20 ^c^	1.51 ± 0.06 ^a^	23.26 ± 0.34 ^b^
	Hôrka	151.27 ± 10.77 ^c^	513.07 ± 18.87 ^d^	12.68 ± 0.07 ^c^	13.91 ± 0.11 ^a^
	Miľpoš	113.77 ± 6.17 ^b^	304.59 ± 19.58 ^c^	6.70 ± 0.03 ^b^	23.27 ± 0.15 ^b^
	Kamienka	46.52 ± 2.07 ^a^	161.52 ± 11.97 ^a^	16.55 ± 0.09 ^d^	22.07 ± 0.17 ^b^
	Lačnov	108.32 ± 8.11 ^b^	242.65 ± 10.32 ^b^	2.01 ± 0.02 ^a^	23.84 ± 0.14 ^b^
**Years**	**Locality**	**NCB**	**CB**	**NCB**	**CB**
2013	Zbojné	76.62 ± 1.91 ^b^	138.27 ± 7.37 ^a^	3.20 ± 0.12 ^a^	21.45 ± 0.20 ^c^
	Hôrka	60.49 ± 4.47 ^a^	163.76 ± 7.23 ^b^	4.70 ± 0.56 ^b^	21.37 0.11 ^c^
	Miľpoš	232.66 ± 7.82 ^e^	1178.98 ± 53.08 ^d^	3.75 ± 0.08 ^a^	14.83 ± 0.13 ^a^
	Kamienka	92.07 ± 2.45 ^c^	542.52 ± 21.37 ^c^	6.44 ± 0.21 ^c^	18.59 ± 0.19 ^b^
	Lačnov	155.54 ± 9.42 ^d^	547.90 ± 20.82 ^c^	11.88 ± 0.25 ^d^	19.57 ± 0.10 ^b^
**Years**	**Locality**	**NCB**	**CB**	**NCB**	**CB**
2014	Zbojné	54.17 ± 4.12 ^b^	238.37 ± 7.74 ^c^	3.07 ± 0.04 ^b^	21.20 ± 0.12 ^b^
	Hôrka	45.96 ± 3.75 ^a^	190.54 ± 5.96 ^a^	1.39 ± 0.01 ^a^	22.69 ± 0.33 ^c^
	Miľpoš	59.74 ± 3.72 ^b^	281.11 ± 14.50 ^e^	1.87 ± 0.12 ^a^	18.47 ± 0.29 ^a^
	Kamienka	79.58 ± 2.85 ^c^	202.87 ± 14.91 ^b^	6.70 ± 0.09 ^d^	23.09 ± 0.44 ^c^
	Lačnov	272.94 ± 9.00 ^d^	266.32 ± 17.97 ^d^	4.24 ± 0.02 ^c^	21.02 ± 0.42 ^b^

^A^NCB–non-crushed berries; ^B^CB–crushed berries; ^C^DM–dry mass; Statistical differences (ANOVA, *p* < 0.001) between localities within each year are marked with different letters (^a,b,c,d,e^).

**Table 3 molecules-27-09027-t003:** Correlations between localities, year, phenols content, FRAP and terpene.

	* Pearson Phenols/FRAP		* Pearson Phenols/Terpene
**Year**	**NCB**	**CB**	**CB**
2012	−0.531	0.843	0.504
**Year**	**NCB**	**CB**	**CB**
2013	−0.349	0.742	−0.684
**Year**	**NCB**	**CB**	**CB**
2014	−0.224	0.617	−0.220

* Correlation coefficient according to Pearson.

**Table 4 molecules-27-09027-t004:** The changes of the sum of precipitation and average temperature in observed localities within three years of research.

Year	Locality (Meteo Station)	Precipitation	Temperature
		Sum for 12 Months (mm)	Average for 12 Months (°C)
2012	Zbojné (Medzilaborce)	808	8.3
	Hôrka (Poprad)	550	6.9
	Miľpoš, Lačnov (Plaveč)	721	7.1
	Kamienka (Podolínec	685	7.3
	**Average**	**691**	**7.4**
2013	Zbojné (Medzilaborce)	815	8.6
	Hôrka (Poprad)	676	7.0
	Miľpoš, Lačnov (Plaveč)	761	7.4
	Kamienka (Podolínec	704	7.5
	Average	739	7.6
2014	Zbojné (Medzilaborce)	752	9.5
	Hôrka (Poprad)	774	8.2
	Miľpoš, Lačnov (Plaveč)	880	8.9
	Kamienka (Podolínec	836	8.5
	**Average**	**810.5**	**8.8**

## Data Availability

Not applicable.

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
