# Peer review of "Seasonal Variability of *Juniperus communis* L. Berry Ethanol Extracts: 2. In Vitro Ferric Reducing Ability of Plasma (FRAP) Assay"

_molecules, 2022, doi:10.3390/molecules27249027_

Round 1

Reviewer 1 Report

Recommendation: Major

The manuscript Seasonal Variability of Juniperus communis L. Berry Ethanol Extracts In Vitro Ferric reducing ability of plasma (FRAP) assay, the methodology was reasonable and technically sound.

Some weak points of the study in my opinion;

-No principal component analysis and cluster analysis

-Juniper berries were harvested in 2012-2014, how were they preserved or when these analyzes were performed. Because how should we be convinced of the reliability of the analysis results until today's date.

-There is no analysis or detail analysis on a component basis. such as LC/MS, HPLC, GC

My comment to the current version of the article is below

Point 1. I suggest you review the title.

Point 2 In the abstract, use numerical expressions to support your results.

Point 3. The phrase "statistically significantly higher activity" should be changed.

Point 4. I recommend that figure 1 and figure 2 be prepared more professionally. Add axis descriptions.

Point 5. How are non-crushed juniper berries (NCB) and crushed juniper berries (CB) prepared? state clearly

Point 6. Statgraphics 5.0 Please specify company information

Point 7. Write the explanations of the abbreviations under Tables 1,2,3

Point 7. Make suggestions for future studies in the conclusion section.

Author Response

Response to Reviewer 1 Comments

Dear Reviewer,

Thank you very much for your time to revie and suggest improvement to our manuscript. Our respond to your comments are below.

The manuscript Seasonal Variability of Juniperus communis L. Berry Ethanol Extracts In Vitro Ferric reducing ability of plasma (FRAP) assay, the methodology was reasonable and technically sound.

Some weak points of the study in my opinion;

-No principal component analysis and cluster analysis

Authors: There was provided analysis of total phenols in NBC and CB extracts and terpene hydrocarbons in CB extracts. In previous publication was published content and composition of the essential oils from the juniperus galbuls fromthe same samples (Fejér, J.; Gruľová, D.; Eliašová, A.; Kron, I.; De Feo, V., 2018. Influence of environmental factors on content and composition of essential oil from common juniper ripe berry cones (Juniperus communis L.). In: Plant Biosystems. Vol. 152, No. 6, 2018, p. 1227-1235).

-Juniper berries were harvested in 2012-2014, how were they preserved or when these analyzes were performed. Because how should we be convinced of the reliability of the analysis results until today's date.

Authors: The galbuls of the juniperus were dried after harvest and stored in cooler. Extracts were prepared in 2016 and directly after extract preparation, there were provided FRAP method for antioxidant activity of the extracts. There was no time to publish this data earlier.

-There is no analysis or detail analysis on a component basis. such as LC/MS, HPLC, GC

Authors: There was provided analysis of total phenol and GCMS analysis of terpene hydrocarbons

My comment to the current version of the article is below

Point 1. I suggest you review the title.

Authors: It is a second publication of the series about antioxidant activity. First publication was published in 2020 as Fejér, J.; Kron, I.; Gruľová, D.; Eliašová, A. Seasonal Variability of Juniperus communis L. Berry Ethanol Extracts: 1. In Vitro Hydroxyl Radical Scavenging Activity. Molecules 2020, 25, 4114.

Point 2 In the abstract, use numerical expressions to support your results.

Authors: Some numerical expressions were added in Abstract – marked in yellow

Point 3. The phrase "statistically significantly higher activity" should be changed.

Authors: The sentences were re-phrased as „Statistics by ANOVA confirmed that CB extracts prepared from berries in the year 2013 showed significantly higher activity compared to CB extracts from berries from the years 2012 and 2014.

Point 4. I recommend that figure 1 and figure 2 be prepared more professionally. Add axis descriptions.

Authors: Figure 1 and 2 were modified

Point 5. How are non-crushed juniper berries (NCB) and crushed juniper berries (CB) prepared? state clearly

Authors: The sentences in chapter 4.2 were rephrased for clear understanding

Point 6. Statgraphics 5.0 Please specify company information

Authors: added in chapter 4.6

Point 7. Write the explanations of the abbreviations under Tables 1,2,3

Authors: Abbreviations and explanations were added

Point 7. Make suggestions for future studies in the conclusion section.

Authors: the suggestions were added to conclusion as: „ The study shows that there are natural products, which are safe for practical use for example in food industry. Our finding shall be considered in the looking for new products which will prolong quality of specific food products or in prolonging their shelf life.„

Reviewer 2 Report

There are many studies related to the subject of the current study but antioxidant and atimicrobial capacity of the plants are still popular. there are some comments to be addressed. After that, the manuscript can be accepted.

Specific comments:

The objectives of the paper are not well documented consistently to represent the whole idea so please reconstruct the objectives of the paper at the end of introduction section again.

Methods especially, FRAP can be briefly described.

Statisticals analyses should be given in discussion section in detail.

what is the core messaje for food or other industries?

Author Response

Response to Reviewer 2 Comments

Dear Reviewer.

Thank you very much for your time and suggestion to improve our manuscript.

There are many studies related to the subject of the current study but antioxidant and atimicrobial capacity of the plants are still popular. there are some comments to be addressed. After that, the manuscript can be accepted.

Specific comments:

The objectives of the paper are not well documented consistently to represent the whole idea so please reconstruct the objectives of the paper at the end of introduction section again.

Authors: The objectives of the study were added in introduction as: “It is well known that the environmental conditions have influence on the development of the natural components in plant matrices. The aim of the study is to compare the quality and the potential antioxidant activity of the extracts in different localities and different years.”

Methods especially, FRAP can be briefly described.

Authors:Brief description was added into chapter 4.5

Statisticals analyses should be given in discussion section in detail.

Authors: In this case, our statistical analysis are for the confirmation of the differences in our results. The statistical procedure of itself is not necessary to discuss. Most important are comparison of the result with the other publications.

what is the core messaje for food or other industries?

Authors: the message were added to conclusion as: „ The study shows that there are natural products, which are safe for practical use for example in food industry. Our finding shall be considered in the looking for new products which will prolong quality of specific food products or in prolonging their shelf life.„

Round 2

Reviewer 1 Report

The authors have made the necessary revisions.